# Integrative Genomic and Transcriptomic Profiling Reveals a Differential Molecular Signature in Uterine Leiomyoma versus Leiomyosarcoma

**DOI:** 10.3390/ijms23042190

**Published:** 2022-02-16

**Authors:** Alba Machado-Lopez, Roberto Alonso, Victor Lago, Jorge Jimenez-Almazan, Marta Garcia, Javier Monleon, Susana Lopez, Francisco Barcelo, Amparo Torroba, Sebastian Ortiz, Santiago Domingo, Carlos Simon, Aymara Mas

**Affiliations:** 1Igenomix Foundation, INCLIVA Biomedical Research Institute, 46980 Valencia, Spain; alba.machado@igenomix.com (A.M.-L.); roberto.alonso@igenomix.com (R.A.); 2Research and Development Department, Igenomix SL, 46980 Paterna, Spain; jorge.jimenez@igenomix.com (J.J.-A.); marta.garcia@igenomix.com (M.G.); 3Gynecologic Oncology Department, University Hospital La Fe, 46026 Valencia, Spain; victor.lago.leal@hotmail.com (V.L.); santiago.domingo.delpozo@gmail.com (S.D.); 4Department of Obstetrics and Gynecology, Hospital Universitario La Fe, 46026 Valencia, Spain; monlesancho@gmail.com; 5Department of Pathology, Hospital Universitario La Fe, 46026 Valencia, Spain; susana.lopezagullo@hotmail.com; 6Department of Gynecology and Obstetrics, Gynecology Oncology Unit, Hospital Universitario Virgen de la Arrixaca, 30120 Murcia, Spain; fjbarcelovalcarcel@hotmail.com; 7Pathology Service, Hospital Clínico Universitario Virgen de la Arrixaca, 30120 Murcia, Spain; mariaa.torroba@carm.es; 8Department of Pathology, Complejo Hospitalario de Cartagena, 30202 Murcia, Spain; sortizreina@yahoo.es; 9Department of Obstetrics and Gynecology, Universidad de Valencia, 46010 Valencia, Spain; 10Department of Obstetrics and Gynecology, BIDMC, Harvard University, Boston, MA 02215, USA

**Keywords:** leiomyoma, leiomyosarcoma, exome/transcriptome, mutational pattern, differential gene expression, integrative analysis, diagnostic/prognostic biomarkers, machine learning, classification model

## Abstract

The absence of standardized molecular profiling to differentiate uterine leiomyosarcomas versus leiomyomas represents a current diagnostic challenge. In this study, we aimed to search for a differential molecular signature for these myometrial tumors based on artificial intelligence. For this purpose, differential exome and transcriptome-wide research was performed on histologically confirmed leiomyomas (*n* = 52) and leiomyosarcomas (*n* = 44) to elucidate differences between and within these two entities. We identified a significantly higher tumor mutation burden in leiomyosarcomas vs. leiomyomas in terms of somatic single-nucleotide variants (171,863 vs. 81,152), indels (9491 vs. 4098), and copy number variants (8390 vs. 5376). Further, we discovered alterations in specific copy number variant regions that affect the expression of some tumor suppressor genes. A transcriptomic analysis revealed 489 differentially expressed genes between these two conditions, as well as structural rearrangements targeting ATRX and RAD51B. These results allowed us to develop a machine learning approach based on 19 differentially expressed genes that differentiate both tumor types with high sensitivity and specificity. Our findings provide a novel molecular signature for the diagnosis of leiomyoma and leiomyosarcoma, which could be helpful to complement the current morphological and immunohistochemical diagnosis and may lay the foundation for the future evaluation of malignancy risk.

## 1. Introduction

Uterine leiomyomas (LM) are benign tumors arising in the smooth muscle cells of the uterine wall. They are the most common pelvic tumors in women, with a prevalence of >80% for African American and ~70% for Caucasian women before 50 years of age [1]. Although LM are non-malignant tumors, the risk of hidden undiagnosed malignancy, such as leiomyosarcoma (LMS), occurs in one among 498 uterine tumors [2,3,4].

Histological diagnosis is the gold standard option for myometrial tumors [5,6]. However, LM and LMS share clinical symptoms and morphological features [7,8], sometimes hindering their differential diagnosis and introducing the risk of the future potential spread of undiagnosed LMS with the use of power morcellators [9]. Besides, alternative invasive approaches, such as laparotomy-based procedures, increase morbidity, mortality, and cost for the patient and healthcare system [10]. 

Despite LM and LMS having been previously characterized at a molecular level [11,12,13], the differential profiling of these myometrial tumors based on genomic/transcriptomic characteristics could allow us a better understanding of the underlying tumorigenic processes as well as to develop novel tools that may aid the current clinical diagnosis. Given these challenges, we aim to discover specific molecular signatures for the differential diagnosis of myometrial tumors. 

In this study, we identified that LM and LMS have significant mutational heterogeneity and differences in copy number alterations at the DNA level, while a specific transcriptomic profile and multiple structural rearrangements were detected at the RNA level. With these data, an integrated molecular analysis was performed to assess the effect of copy number variants (CNVs) on gene expression. Targeted RNAseq data and artificial intelligence were used to create a predictive model for the comprehensive molecular classification of LMS and LM at the tumor-tissue level.

## 2. Results

### 2.1. Clinical Study Design

After obtaining informed consent from the eligible patients, tumor samples were collected from women undergoing a hysterectomy or myomectomy as a surgical treatment for a primary myometrial tumor. Following histological diagnoses according to WHO criteria [14], a total of 106 selected LM and LMS samples were separated in two cohorts, the experimental cohort and the validation cohort (Appendix A). The epidemiological, histopathological, and clinical outcomes of the patients involved in the study are summarized in Table 1. 

### 2.2. Identification of Differential Somatic Single Nucleotide Variants and Insertions/Deletions

We performed whole-exome sequencing in 44 LM and 34 LMS tumors to screen for single nucleotide variants (SNVs) and insertions/deletions (indels). We detected 181,354 small variants in the LMS tumors, of which 171,863 were SNVs and 9491 were small indels. In the LM samples, we detected 85,250 small variants, of which 81,152 were SNVs and 4098 were small indels. Among these variants, 27.63% where shared between LMS and LM, while 34.56% were LMS-exclusive and 37.81% were LM-exclusive (in at least one sample).

Although comparative analyses of SNVs showed a similar distribution, LMS had a higher mean number of alterations per sample compared to LM (*p* < 0.05). Then, we focused on group-exclusive variants present in at least six samples of each group to find exclusive variants involved in the pathogenic process of each group. As a result, we found a total of thirteen variants affecting eight different genes exclusive to LMS (Figure 1A), while we found a total of twelve mutations affecting twelve genes exclusive to LM (Figure 1B). In both groups, these mutations were mostly missense mutations, although, in the LMS group, there were other types of variants, i.e., in frame indels affecting the *IQCJ*-*SCHIP1* gene, structural interaction variants affecting the *GAPDH* gene, and frameshift variants affecting the *EEF2* gene. Interestingly, most of the variants detected have been previously reported in other cancer types in the COSMIC database (Appendix A).

Using this SNV information, we compared the mutational spectrum for the LM and LMS tumors through the relative contribution of six base substitution types (Figure 1C), which were then decomposed into two distinct mutational signatures (Figure 1D). 

In an attempt to relate our findings to known mutational signatures, we searched in the COSMIC database [15] and identified four (1, 5, 12, and 20) out of 30 existing signatures (Figure 1E). Among them, signature 1 results from an endogenous mutational process initiated by spontaneous deamination of 5-methylcytosine, while signature 5 exhibits transcriptional strand-bias for T > C substitutions at ApTpN context. We also identified signature 20, which is associated with defective DNA mismatch repair due to high numbers of small indels at mono/polynucleotide repeats. While these signatures have already been identified across 40 different human cancer types, signature 12 represents a novel mutational signature only present in these uterine tumors, showing similarities to liver cancer and exhibiting a strong transcriptional strand-bias for T > C substitutions as additional mutational features (Figure 1E). 

Since we detected a molecular phenotype with a defect in the DNA mismatch repair system, we next evaluated microsatellite instability (MSI) status to predict the outcome in LM and LMS tumors, although no differences were found in the number of alleles or the fragment size (Appendix A).

### 2.3. Identification of Copy Number Variants

We next compared the somatic copy number variants (CNVs) in LMS and LM. A total of 14,467 CNVs were detected in LM, while 14,950 CNVs were detected in LMS. Despite the similar results, Student’s *t*-test showed a significant difference in the mean values of CNVs per sample between LMS (439.7) and LM (328.8) (*p* = 5.61 × 10^−5^). Because some CNVs were present in more than one sample within each group, we filtered the unique CNVs per group, obtaining a total of 8390 CNVs in LMS and 5376 CNVs in LM. In terms of their structural nature, 18.2% of the CNVs in LMS were deletions, while 73.1% were duplications. Specifically, 3.5% were LMS-specific deletions present in more than one sample, and 5.2% were LMS-specific duplications present in more than one sample. In the LM group, 11.7% of the CNVs were deletions and 84.5% were duplications, with only 0.1% tumor-specific deletions and 3.6% tumor-specific duplications (Figure 2A). While the CNV profile for LMS was heterogeneous and showed alterations in most chromosomes, the LM tumors had recurrent losses in chromosomes 1, 13, 14, 15, and 22 and recurrent gains in chromosomes 12 and 19 (Figure 2B). 

Kaplan–Meier survival curves were generated to assess the association between LMS-specific CNVs and clinical prognosis based on overall survival. We selected 12 of the most frequent LMS-specific CNVs present in at least 10 out of 34 LMS (Appendix A) and found statistically significant differences between the patients with disruptions in at least 67% of the CNVs. Remarkably, we observed a tendency where these patients with aberrant CNVs had shorter survival times than those with normal copy number values (diploid) in these regions (Figure 2C). To account for possible confounding factors, we performed a multivariate survival analysis, although, as expected, the results were non-significant, probably due to the low sample size and the high number of variables included. Still, given the potential clinical relevance of this finding, unsupervised hierarchical clustering using the 370 genes included in these CNVs demonstrated a clear separation between the LMS and LM tumors at the DNA level (Figure 2D). 

### 2.4. Proximal Expression Effects Inferred from Integration with CNVs

Because CNVs can involve a large region containing multiple genes, we integrated the detected CNV regions with exome-wide gene expression data from RNAseq in LM and LMS tumors (Appendix A). As a result, we identified regions in the LMS samples located in chromosomes 5, 11, 14, and 16 in which gains or losses were significantly associated with changes in expression involving five genes (Figure 2E). Specifically, in LMS (Figure 2E, upper), we detected *TRIP13* with a positive correlation between gains and higher expression, while losses were associated with lower expression. However, the expression of other genes increased when there was a loss in the corresponding region (*CDKN1C*) or decreased when there was a gain (*BATF*, *DECR2*, *LUC7L*), possibly due to different mechanisms of regulating gene expression. The same analysis was performed in the LM samples (Figure 2E, lower), showing regions located in chromosomes 5, 6, 7, 9, 11, 14, and 17. As in LMS, some genes showed a positive correlation between copy number state and expression (*ZSCAN9*, *MARK3*, *CHRNB1*, *WRAP53*, *YBX2*), while, in the remaining genes, gains in the chromosomal region were associated with lower expression values due to more complex gene regulation.

### 2.5. Structural Rearrangements Affect Specific Regions and Genes in LM and LMS 

Further, we identified high-confidence fusion transcripts in 29.5% of the LM cases and 61.8% of LMS arising from chromosomal rearrangements, resembling chromothripsis in some cases, such as LMS25 and LMS26 (Figure 3A). 

Chromosomes 3, 8, 11, 13, 17, and X were the most frequently affected in LMS (Figure 3A), while chromosomes 1, 3, 6, and 14 were the most impacted in LM (Figure 3B). Although no recurrent fusions were detected in any tumors, multiple rearrangements targeting the chromatin remodeling protein *ATRX* were identified in LMS, and a known oncogene, *RAD51B*, was identified in LM. Specifically, *ATRX* was fused with several gene partners, such as *TRAPPC9*, *RP11-56A10.1*, and *EZH1*, resulting in a non-functional fusion protein lacking the helicase ATP-binding domain and/or the helicase C-terminal domain in the LMS tumors (Figure 3A). In the LM tumors, *RAD51B* was fused with *HMGA2*, *NCOR2*, and *NUDT3*, indicating the potential of these fusions to drive tumorigenesis (Figure 3B). 

### 2.6. Differential Transcriptomic Characterization of LMS versus LM 

We next sought to identify differential gene expression footprints by the RNAseq analysis of 44 LM and 34 LMS tumors. A class comparison detected a total of 489 DEGs, 416 significantly upregulated and 73 downregulated, between LMS and LM (FDR < 0.05 and |logFC| > 2). Some of the most significant upregulated genes in LMS were validated by RT-PCR, confirming significant overexpression and correlation between qPCR and RNAseq (Appendix A). 

Next, unsupervised hierarchical clustering grouped the LMS samples in a homogeneous cluster of 29 samples, while 30 LM samples were detected in a separate cluster. Of note, another cluster included the remaining LM with some LMS (LMS03, LMS11, LMS26, LMS35, and LMS62) (Figure 4A). Based on the heatmap/dendrogram, these LMS samples appeared closer to the LM group, suggesting that their molecular profile was more similar to the LM samples than the LMS samples. This was in line with clinical information of the corresponding patients as only one out of the five LMS patients died due to the disease, while the other four are still alive, reinforcing that these tumors may have intermediate characteristics but are closer to LM.

To better understand the molecular functions, biological processes, and pathways differentially regulated by the 489 DEGs, we performed Kyoto Encyclopaedia of Genes and Genomes (KEGG), Gene Ontology (GO), and Reactome enrichment analyses. In total, we detected 10 KEGG pathways, 83 GO terms, and 92 Reactome pathways (Appendix A). Briefly, the 489 DEGs were mostly involved in cell cycle-associated processes (e.g., nuclear division, chromosome segregation, regulation of cell cycle phase transition, meiosis) according to all the databases (Appendix A). In summary, these results indicate that differences in the expression between LMS and LM occurred in genes associated mainly with cell cycle and nucleic acid metabolism, suggesting that alterations of these processes occur differently and have different consequences for each tumor type. 

### 2.7. Model Creation and Validation for Differential Molecular Diagnosis of LMS and LM 

To classify LM and LMS, we developed a machine learning approach based on the transcriptomic signatures of each group since we found that management, analysis, and biological comprehension were more straightforward when using RNAseq data. After feature pruning, the final model was composed of 19 DEGs and was able to correctly classify all the samples in the validation set.

Based on this model, we built a targeted sequencing panel using AmpliSeq technology for the 19 selected genes, which was used to re-analyze all the previous LM and LMS tumors (*n* = 44 and *n* = 32, respectively, since two LMS samples were filtered due to poor sequencing quality) in addition to new samples (eight LM and ten LMS). 

Next, the total 96 samples were randomly split into a training set to build the machine learning model and a test set to validate the model (75% and 25% class-balanced samples for the training and test sets, respectively). Specifically, the gradient boosting algorithm was used to build a new model, which achieved optimal values of sensitivity and specificity since it was able to correctly classify all the test samples. The feature selection resulted in a final model that consisted only of 19 genes, out of which only three, *COL4A5*, *MFAP5*, and *ITGA9*, were overexpressed in leiomyoma, while the rest were overexpressed in leiomyosarcoma). The unsupervised clustering of the samples based on these 19 genes allowed separation through a heatmap of two distinguishable groups of LM and LMS. However, and resembling results from previous RNAseq analysis, two LMS (LMS26 and LMS39) were clustered in a group opposite to those confirmed by pathology (Figure 4B).

Further, the model was used to classify the samples and to calculate class probabilities, allowing a more fine-tuned classification of the samples, where we defined a “warning range” for those tumors where the model was not confident enough, defined as probabilities of <75% for each group (Figure 4C). Interestingly, this model could correctly classify all the samples with high class probabilities, even for sample LMS39 with the lowest LMS probability.

## 3. Discussion

The search for molecular criteria to differentiate uterine myometrial tumors represents an important current diagnostic challenge, where molecular profiling could be a powerful complement to current diagnosis based on the clinical presentation, imaging features, and microscopic morphologic characteristics. Since our main aim was to build a preliminary differential diagnostic tool, we have focused on the comparison between the most prevalent benign myometrial tumors, LM, and the most aggressive ones, LMS. 

While we are aware that excluding other tumor types, such as STUMPs, inflammatory myofibroblastic tumors (IMT), or undifferentiated uterine sarcomas, is a relevant limitation of our study, we decided to discard these samples in order to avoid introducing noise in our molecular analysis, also considering that, given the low incidence of these tumors, the fact that excluding them would not have a significant impact on our preliminary classification tool. 

Following a similar rationale, we chose to limit our analysis to only conventional LMS and LM given the low prevalence of non-conventional variants and their differences, which may also introduce noise in our analyses. 

Consequently, in this study, exome- and transcriptome-wide analyses were performed on histologically confirmed conventional LM and LMS to investigate molecular differences between and within these two different entities.

In terms of small variants, we found that only a small percentage of variants (27.63%) were shared by the two groups, while the rest were exclusive of LM or LMS, suggesting independent mechanisms for tumorigenesis. Interestingly, we found the c.2443C > T mutation in the *REST* gene involves the loss of the repressor in LM and promotes the aberrant expression of *GPR10* in the PI3K/AKT mTOR pathway [15]. Conversely, in LMS, we detected three different mutations affecting the *OBSCN* gene, which has been proposed as a molecular tool to differentiate LMS from gastrointestinal stromal tumors within a two gene-classifier [16]. Likewise, mutations in *CCDC68* involving microtubule organization have been reported in different cancer types [17,18]. Moreover, with variant information, we detected for the first-time signature 12 of COSMIC as a mutational mark in LM and LMS. While its etiology remains unknown, its presence has been associated with a small percentage (<20%) of mutations in liver cancer [16]. 

Additionally, we found differences in the number and type of somatic CNVs, suggesting that they could be used for the differential diagnosis of myometrial tumors. Furthermore, we provided novel insights into the potential usefulness of CNVs as a prognostic biomarker for patients with LMS. However, the low number of samples available for the survival analysis (mainly due to a lack of data resulting from the use of commercial samples or censoring) is a strong limitation, and, thus, the tendency we observed should be carefully interpreted and further explored and validated.

Because CNVs may cause changes in transcription levels associated with cancer [17], we performed an integrative analysis on the CNV and expression data. Interestingly, we found that alterations in specific CNV regions in LMS affected the expression of *CDKN1C*, a cyclin/Cdk inhibitor that may act as a tumor suppressor [18]. 

Further, we screened transcriptomic data to detect fusions in LM and LMS. While no recurrent fusions were detected in LMS or LM, multiple rearrangements targeted *ATRX* and *RAD51B*, respectively. The loss of expression of *ATRX* has been associated to the Alternative Lengthening of Telomeres phenotype, which allows tumor cells to escape programmed cell death. In uterine LMS, this mechanism has been associated with a poor prognosis and overall survival [19,20]. Additionally, we identified the *RAD51B*–*HMGA2* fusion transcript in LM, which may have a role in tumor progression, as previously described [21].

The transcriptomic results revealed 489 differentially expressed in LMS versus LM. Although some of these genes have been reported [11,22,23], we identified specific DEGs involved in mitotic spindle checkpoint regulation, including *AURKA*, *SPAG5*, *NUF2*, *BUB1B*, and *KIF14* [24], while the *CCDC68* gene affected by a deleterious variant mentioned in Section 2.1 is also involved in this process. The defective segregation of chromosomes and microtubule–kinetochore–spindle formation cause tumor cells to become aneuploid, allowing DNA-damaged cells to skip the spindle assembly checkpoint, suggesting a possible molecular mechanism for the development of LMS. One of the most significant genes was *AURKA*, a cell cycle protein that is also overexpressed in cervical [25] and ovarian [26] cancers and seems to be key in the pathogenesis of LMS since its inhibition results in cell cycle arrest and apoptosis in LMS cell lines [27]. 

Further analysis based on hierarchical clustering revealed some LMS samples that unexpectedly clustered with LM samples. Accordingly, after reviewing the clinical features and follow-up, we found that most of these patients exhibit higher overall survivals than reported by previous retrospective studies [28]. These findings are consistent with our prior premise that myometrial tumors can be differentiated based on their transcriptomic profiles.

We finally built a classification model composed of nineteen genes, sixteen overexpressed in LMS and three overexpressed in LM, possibly suggesting that the overexpression of a small subset of genes results in a more disrupted molecular profile that is sufficient to indicate that a tumor is more likely to be a malignant uterine leiomyosarcoma. The genes in this panel are involved in different processes and functions, including DNA replication (*BRCA2*, *CHAF1A*, *E2F7*, and *EXO1*); DNA damage repair (*ARHGAP11A* and *PBK*); extracellular matrix formation and interaction (*COL4A5*, *ITGA9*, and *MFAP5*), and, lastly, segregation of chromosomes and microtubule–kinetochore–spindle formation (*CCDC34*, *CDCA5*, *CENPE*, *CENPF*, *CENPH*, and *MLF1IP*), which, as we have mentioned, could be an affected pathway in the tumorigenesis of LMS. As a result, our model could be a representation of the affected pathways that differ between LMS and LM, where the genes associated with alterations in the extracellular matrix suggest the diagnosis of LM (since *COL4A5*, *ITGA9,* and *MFAP5* are the only genes overexpressed in LM), and the genes associated with defective DNA replication or DNA damage repair or segregation of chromosomes suggest the diagnosis of LMS. 

Moreover, this model correctly classified all the samples in the validation set with high sensitivity and specificity, while the class probabilities calculated for each sample showed the predictions of the model were of high confidence (all >75% for their class). It is also of note that, even though the unsupervised hierarchical clustering of global RNAseq showed a mixed cluster of LMS and LM samples, the machine learning algorithm effectively classified these samples, once again demonstrating the potential of the predictive tool. 

Nevertheless, although promising, the genomic and transcriptomic outcomes are based on the analysis of the entire tumor, which implies a technical limitation due to intratumoral heterogeneity. Additional studies with larger numbers of samples are also needed to validate and clinically use this tumor classification model. 

In summary, our findings provide a novel molecular profile and candidate gene targets to discriminate LM from LMS at the tumor-tissue level, establishing a potential diagnostic tool that could be helpful to complement morphological and immunohistochemical diagnostic features. 

Accordingly, this molecular-driven test linked to functional histology might provide a framework for objective and specific diagnosis at the tumor tissue level. The challenge, however, for both the pathologist and the physician, is how best to effectively integrate this morphological and molecular information into a comprehensive diagnosis and treatment plan.

## 4. Materials and Methods

Detailed description of the materials and methods used in this study are provided in the Appendix A and Methods (Appendix B). 

### 4.1. Clinical Sample Collection

Use of human tissue samples was previously approved by the IRB of the hospitals involved: Hospital La Fe, Valencia, Spain (24 July 2019), Hospital Virgen de la Arrixaca Murcia, Spain (17 December 2019), Hospital Santa Lucía Murcia, Spain (28 January 2020), and Fundación Instituto Valenciano de Oncologia (IVO), Valencia, Spain (2 December 2020). All patients signed and provided written informed consent. 

Briefly, formalin-fixed paraffin-embedded (FFPE) tumor samples were collected from 119 women undergoing hysterectomy or myomectomy as surgical treatment for primary myometrial tumors (Appendix A). Before further processing, anonymized samples were evaluated by two pathologists with wide experience in gynecology, who histologically confirmed a diagnosis of LM or LMS according to WHO criteria [14] and provided us with paraffin blocks consisting of at least 85% of tumor tissue. Patients with other gynecological tumors, disorders, malignancies, or diagnosed bacterial, fungal, or viral infections were excluded (*n* = 16). 

Specifically, Hospital la Fe contributed with 56 LM samples and 13 LMS tumors. Hospital Santa Lucía provided to this study 6 LMS, 12 LMS came through Hospital Virgen de la Arrixaca, while 3 LMS were provided by IVO, and 13 LMS were supplied by Origene Technologies Inc. (Rockville, MD, USA). Selected LM and LMS samples were then split into two cohorts: the experimental cohort, which was used to study global DNA and RNAseq profiles (44 LM, 34 LMS), and the validation cohort, including additional new samples (8 LM, 10 LMS) to perform targeted sequencing and model validation (Appendix A). Epidemiological, histopathological, and clinical outcomes are summarized in Table 1. This study was registered on ClinicalTrials.gov (ID NCT04214457), and data were monitored by a clinical research associate. 

### 4.2. DNA Sequencing and Analysis

After nucleic acid isolation, DNA libraries were constructed using the KAPA Hyper Prep kit (Roche, Basel, Switzerland) and enriched using a panel of 571 hematological-associated genes. DNA sequence data were demultiplexed and aligned to the human hg19 genome (CRGCh37) using BWA [29] and SAMtools [30] following the quality control and metrics detailed in Appendix A. 

Sequencing data were analyzed for small variants using Freebayes (https://github.com/freebayes/freebayes, accessed on 31 December 2021) and annotated with SnpEff (https://github.com/pcingola/SnpEff, accessed on 31 December 2021). Additional assessment of somatic mutational signatures inferred from SNVs was done using MutationalPatterns [31]. Further analysis for microsatellite instability (MSI) was performed using a set of six mononucleotide repeat markers (Appendix A) following published protocols [32]. Lastly, we used CNVkit with default parameters (https://github.com/etal/cnvkit, accessed on 31 December 2021), for CNV detection. 

### 4.3. RNA Sequencing and Analysis

RNA libraries from 44 LM and 34 LMS were constructed using Truseq RNA exome (Illumina, San Diego, CA, USA) and aligned to the human hg19 genome using STAR (https://github.com/alexdobin/STAR, accessed on 31 December 2021) (Appendix A) to estimate gene transcript abundance with HTseq (https://github.com/simon-anders/htseq, accessed on 31 December 2021). 

First, arriba (https://github.com/suhrig/arriba/, accessed on 31 December 2021) was used for detection of fusions on RNAseq data, which were then validated by immunohistochemistry. Subsequently, differential expression analysis between LMS and LM was performed using edgeR [33] and subjected to functional analysis. Some of the most significant DEGs were finally validated using RT-qPCR (Appendix A). 

### 4.4. Integrative DNA/RNA Analysis

To evaluate the association between CNVs and RNAseq counts, we used CNVRanger (https://github.com/waldronlab/CNVRanger, accessed on 31 December 2021), excluding genes with <20 counts per million (cpm) and CNV regions with <10 samples in a group deviating from 2n, with a 1 Mbp window and *p-value* < 0.01 (Appendix A).

### 4.5. Building and Validating the Classification Model 

Since we found that analysis and clinical interpretation were more straightforward when using RNAseq, we built a classification model with this data using caret package (https://github.com/topepo/caret, accessed on 31 December 2021). For this purpose, our sample cohort was randomly split, keeping balanced class distributions into a training (75% of samples) and test set (25% of samples). We used the Adaboost algorithm on DEGs cpm to perform a prior selection of predictor genes.

Validation of the model was next performed by re-sequencing LM (*n* = 44) and LMS (*n* = 34) samples, adding a new set of 8 LM and 10 LMS samples. Briefly, RNA from 96 FFPE tissue sections was used to prepare libraries with a PCR/amplicon-based workflow (AmpliSeq Library Plus, Illumina, San Diego, CA, USA).

Normalized coverage values were introduced in caret using the gradient boosting algorithm [34]. Once the model was built, the test set was used to construct receiver operating characteristic curves (ROCs), also assessing sensitivity and specificity (Appendix A).

### 4.6. Statistical Analyses

Statistical analyses were performed using R (http://www.R-project.org, accessed on 31 December 2021). Two-tailed Student’s *t*-tests were used to compare quantitative clinical variables in LMS and LM patients and for gene validation using RT-qPCR. All survival analyses were achieved by Cox regression in a multivariate model to account for LMS prognostic factors and compared between groups using the log-rank test. Correlation between RNAseq and qPCR-based logFC was evaluated using Pearson’s correlation test.

## 5. Conclusions

In conclusion, this study provides a novel molecular classification for leiomyoma and leiomyosarcoma tumors and may be a helpful tool for current diagnosis, hopefully laying the foundation for the future evaluation of malignancy risk.

## 6. Patents

A patent disclosure has been filed for the study under the inventors A.M., R.A., and C.S. since 11 October 2021.

## Figures and Tables

**Figure 1 ijms-23-02190-f001:**
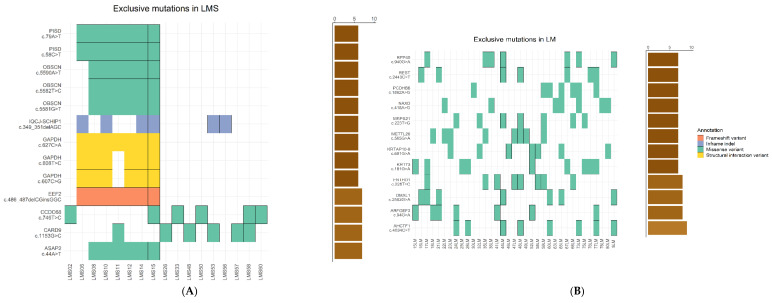
Comparative analysis of single-nucleotide variants (SNVs), insertions/deletions (indels), and mutational signatures for leiomyosarcoma (LMS) and leiomyoma (LM) samples. (**A**) Tumor profile of LMS-exclusive variants, including frequency and type of mutations. (**B**) Tumor profile of LM-exclusive variants, including frequency and type of mutations. In both cases, rows represent individual genes, while columns represent individual tumors. Bars illustrate the number of samples for each exclusive mutation. Types of mutations are annotated according to color. (**C**) Relative contribution of the indicated mutation types to the point mutation spectrum for each tumor type. Error bars indicate standard deviation over all samples. Total number of mutations for LM and LMS is indicated. (**D**) Relative contribution of each indicated trinucleotide changes to the two mutational signatures identified by non-negative matrix factorization (NMF) analysis. (**E**) Heatmap showing relative contribution of each mutational signature described in the COSMIC database for each sample.

**Figure 2 ijms-23-02190-f002:**
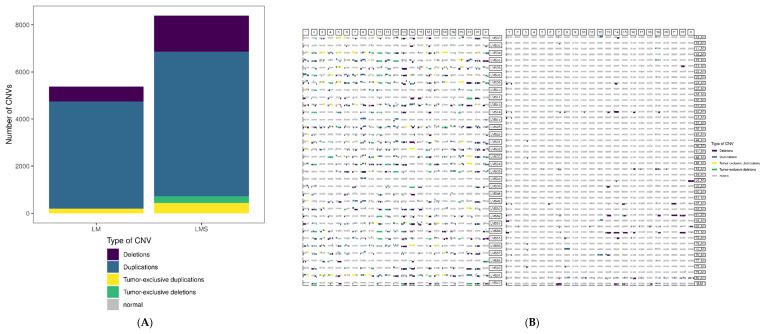
Comparative analysis of copy number variants (CNVs) in leiomyosarcoma (LMS) and leiomyoma (LM) samples and proximal effects from integrative analysis of CNVs and RNAseq data. (**A**) Distribution of CNVs per tumor type. (**B**) Genome-wide CNV distribution in LMS (left) and LM (right). In both cases, rows represent individual samples, while columns represent chromosomes. Types of CNVs are annotated by color, depending on if the deletion/duplication is detected in one sample (purple/blue) or two or more samples (green/yellow). (**C**) Kaplan–Meier plots showing the association between overall survival and alterations in at least 67% of the most frequent CNVs detected in LMS patients. (**D**) Heatmap of unsupervised hierarchical clustering based on the 370 genes affected with the most common CNVs related to patient outcome. (**E**) Proximal effects from the integrative analysis of CNVs and RNAseq data. Boxplots show a region’s expression (*y*-axis, log of normalized counts per million reads mapped) of genes regulated by the specific region (*x*-axis) and colored by copy number state, represented as loss (blue), normal (orange), and gain (red) in LMS (upper) and LM (lower) samples. ** *p*-adjusted value < 0.01; *** *p*-adjusted value < 0.001.

**Figure 3 ijms-23-02190-f003:**
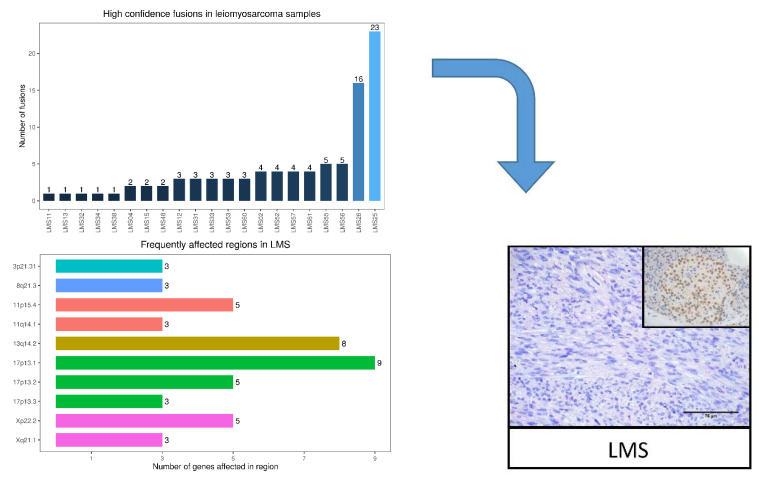
Structural variant plots of chromosomal rearrangements in leiomyoma (LM) and leiomyosarcoma (LMS). (**A**) Bar plot showing the number of high-confidence fusions per LMS sample (upper). Bar plot and ideograms showing the most frequently affected chromosome regions in LMS samples (middle). Schematic representation of the gene sequence and functional protein domain for the most affected gene, *ATRX*, validated by immunohistochemistry (lower), using glioma biopsies as a positive control (right). Scale bar represents 75 µM (*n* = 3). (**B**) Bar plot showing the number of high-confidence fusions per LM sample (upper). Bar plots and ideograms showing the most frequently affected chromosome regions in LM samples (middle). Schematic representation of the gene sequence and functional protein domain for the most affected gene, *RAD51B*, validated by immunohistochemistry (lower) and using gallbladder as a positive control (right). Scale bar represents 75 µM (*n* = 3).

**Figure 4 ijms-23-02190-f004:**
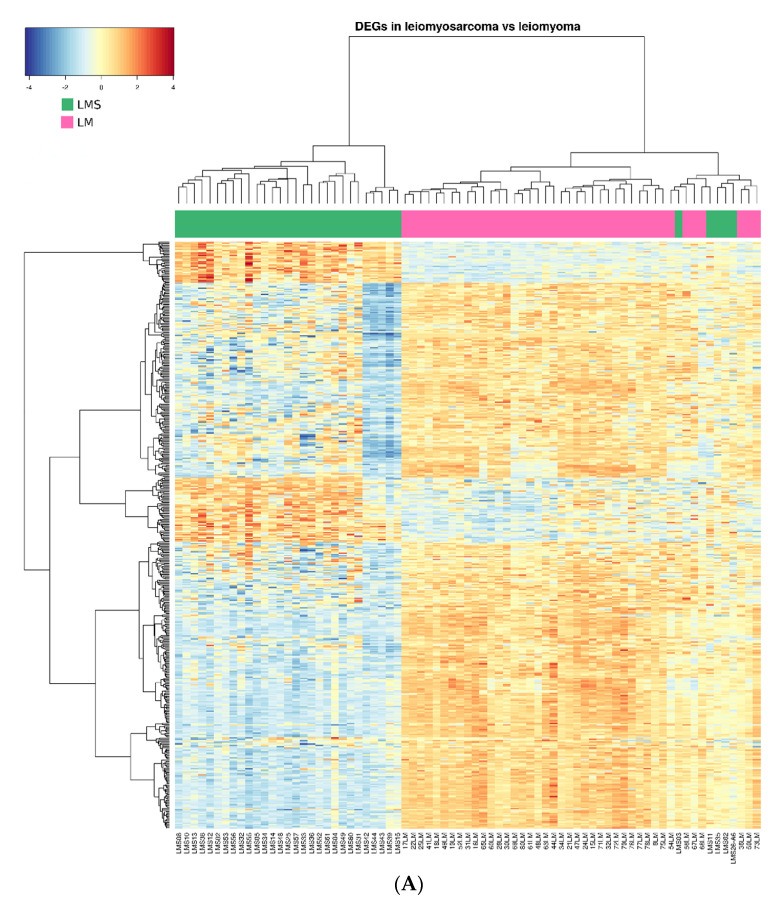
Transcriptional analysis and validation of the targeted gene panel on leiomyoma (LM) and leiomyosarcoma (LMS). (**A**) Heatmap of hierarchical clustering. Top dendrogram shows clustering of samples, and left dendrogram shows clustering of all differentially expressed genes. Colors in the heatmap represent gene expression intensities, with blue indicating low expression and red indicating high expression. The bar on top of the heatmap represents the group by color (green = LMS; pink = LM). (**B**) Heatmap showing clustering of samples using the normalized coverage data for each of the 19 genes. (**C**) Class probabilities predicted by the model for the test set, with the “warning range” highlighted in light orange.

**Table 1 ijms-23-02190-t001:** Epidemiological, demographic, and clinicopathological outcomes of 56 patients diagnosed with uterine leiomyoma (LM) and 47 patients with leiomyosarcoma (LMS) from the experimental cohort.

	Characteristics	LMS	LM
Demographic variables	Age		
	≤30 years	-	2 (3.57%)
	31–40 years	7 (14.89%)	17 (30.36%)
	41–50 years	11 (23.41%)	33 (58.93%)
	51–60 years	20 (42.55%)	2 (3.57%)
	≥61 years	9 (19.15%)	-
	Not available (*n*)	-	2 (3.57%)
	Median (years)	53	44
	Range (years	35–75	28–55
	Ethnicity		
	Caucasian	36 (76.59%)	41 (73.21%)
	African American	1(2.13%)	1 (1.79%)
	Latin	4 (8.51%)	6 (10.71%)
	Asian	1(2.13%)	-
	Arabic	1(2.13%)	-
	Not available	4 (8.51%)	8 (14.29%)
	Body mass index (kg/m^2^)		
	Median	27.15	24
	Range	21.5–34.9	18.20–34.63
	Not available (*n*)	21	13
Gynecologic background	Parity		
	Yes	23 (48.94%)	27 (48.21%)
	No	-	1 (1.79%)
	Not available	24 (51.06%)	28 (50.00%)
	Miscarriage		
	Yes	7 (14.89%)	15 (26.79%)
	No	16 (34.05%)	13 (23.21%)
	Not available	24 (51.06%)	28 (50.00%)
	Menopausal status		
	Premenopausal	15 (38.30%)	46 (82.14%)
	Postmenopausal	18 (31.91%)	2 (3.57%)
	Not available	14 (29.79%)	8 (14.29%)
Symptoms	Pelvic mass		
	Yes	25 (53.19%)	28 (50.00%)
	No	7 (14.89%)	20 (35.71%)
	Not available	15 (31.92%)	8 (14.29%)
	Abnormal uterine bleeding		
	Yes	17 (36.17%)	26 (46.43%)
	No	11 (23.40%)	21 (37.50%)
	Not available	19 (40.43%)	9 (16.07%)
	Abdominal pain		
	Yes	16 (34.04%)	14 (25.00%)
	No	11 (23.41%)	32 (57.14%)
	Not available	20 (42.55%)	10 (17.86%)
Imaging	CT		
	Yes	14 (29.79%)	7 (12.50%)
	No	16 (34.04%)	42 (75.00%)
	Not available	17 (36.17%)	7 (12.50%)
	MRI		
	Yes	5 (10.64%)	5 (8.93%)
	No	22 (46.81%)	44 (78.57%)
	Not available	20 (42.55%)	7 (12.50%)
	Ultrasound		
	Yes	31 (65.96%)	49 (87.50%)
	No	-	-
	Not available	16 (34.04%)	7 (12.50%)
	Tumor size (cm)		
	Median	13	7.4
	Range	mar-24	0.25–25
	Not available (*n*)	17	10
	Suspected uterine sarcoma		
	Yes	15 (31.91%)	5 (8.93%)
	No	15 (31.91%)	44 (78.57%)
	NA	17 (36.18%)	7 (12.50%)
Surgical treatment	Endometrial biopsy		
	Yes	15 (31.91%)	39 (69.64%)
	No	17 (27.66%)	10 (17.86%)
	Not available	19 (40.43%)	7 (12.50%)
	Primary surgery		
	Laparoscopic hysterectomy	1 (2.13%)	12 (21.43%)
	Laparoscopic myomectomy	-	5 (8.93%)
	Laparotomic hysterectomy	33 (70.21%)	21 (37.50%)
	Laparotomic myomectomy	-	11 (19.64%)
	Not available	13 (27.66%)	7 (12.50%)
Clinical follow-up	Recurrence		
	Yes	19 (40.43%)	49 (87.50%)
	No	9 (19.16%)	-
	Not available	19 (40.43%)	7 (12.50%)
	Status		
	Alive	12 (25.53%)	48 (85.71%)
	Deceased	12 (25.53%)	-
	Not available	23 (48.84%)	8 (14.29%)
	Follow-up (months)		
	Median	24	-
	Range	8–116	-
	Not available (*n*)	29	-

## Data Availability

The data presented in this study are available on request from the corresponding author due to ethical/privacy restrictions.

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
