# Peer review of "Integrative Genomic and Transcriptomic Profiling Reveals a Differential Molecular Signature in Uterine Leiomyoma versus Leiomyosarcoma"

_ijms, 2022, doi:10.3390/ijms23042190_

Round 1

Reviewer 1 Report

This is interesting paper. Study findings  enable to assess molecular profile of the most frequent neoplasm- uterine leiomyoma and differentiate these benign tumors from malignant leiomyosarcoma. Authors noted that specific for both tumors molecular signature may be helpful in the future for the evaluation of malignancy risk and complete current diagnostic tools.

Results section is appropriate described, but the images of Figures 1-4 are not legible enough. Authors should improve the resolution of images.

Supplementary tables, figures and detailed description of the material and methods in Supplementary files are clear and readable.

Tables are clear and the data contained in the table readable.

Material and methods were adequately described.

References are adequate, but newer articles in the subject of differentiation LM and LMS are available, which I suggest to cite (ie: doi: 10.1002/cncr.31754; doi: 10.3390/cells10010053; or doi: 10.1038/s41598-020-77666-y)

Reviewer 2 Report

3rd February, 2022

Review of the Manuscript ID: ijms-1582000, by A. Machado-Lopez et al., entitled: “Integrative genomic and transcriptomic profiling reveals a differential molecular signature in uterine leiomyoma versus leiomyosarcoma” that is intended to be published as the Article in International Journal of Molecular Sciences

(separate Microsoft Word file as Reviewer Attachment for Manuscript ID ijms-1582000 Int. J. Mol. Sci. 3rd February 2022 that includes Comments to the Authors is also uploaded)

Taking into consideration research highlight, contribution of the Authors to the progress in the research area, thorough manner of data presentation, perfectly writing in English, abundance of Materials and Methods and Results, and diligent tabular and graphic documentation, the quality of this paper deserves praise and merits my support. The Authors have received the very high scores from me for the originality, importance of the work and the scientific value of their paper. In my opinion, the current paper provides insightful interpretation of topical and coming trends in comprehensively elaborating the innovative (artificial intelligence-assisted) strategies of differential and molecular signature-based diagnostics that can be applied to properly identify and distinguish the multifaceted etiopathogenesis of such severe oncological uterine disorders as: leiomyosarcoma (LMS) and leiomyoma (LM). This could enable oncologists to not only develop and optimize the novel anti-cancer therapeutic strategies and potent personalized treatments, but also estimate the risks related to malignancy and metastasis of the above-mentioned myometrial tumors in female patients afflicted with these neoplastic diseases arising in utero.

Summing up, I strongly recommend the Editorial Board to allow for publication of this excellent paper in International Journal of Molecular Sciences, after the minor revision of the manuscript will have been completed by the Authors and provided that the Authors are ready to consider all the Reviewer comments indicated below:

1) There is a lack of the separate Conclusions and Abbreviations subsections in the paper. Therefore, these subsections should have been added by the Authors at the end of the manuscript. The Abbreviations subsection should have been prepared in order to comprehensively elucidate and expand a broad spectrum of the in-text abbreviations, which have been used by the Authors in all the subsections of their paper.

2) The References section has to be carefully and thoroughly prepared in the format compatible with the requirements of International Journal of Molecular Sciences.

General Comment of the Reviewer:

Before the manuscript will have been accepted for publication in International Journal of Molecular Sciences, it requires the minor revision (according to all the suggestions and recommendations indicated above by the Reviewer).
